# The role of prefrontal cortex in the control of feature attention in area V4

Narcisse P. Bichot[1]*, Rui Xu[1], Azriel Ghadooshahy[1], Michael L. Williams[1] & Robert Desimone[1]

When searching for an object in a cluttered scene, we can use our memory of the target object features to guide our search, and the responses of neurons in multiple cortical visual areas are enhanced when their receptive field contains a stimulus sharing target object features. Here we tested the role of the ventral prearcuate region (VPA) of prefrontal cortex in the control of feature attention in cortical visual area V4. VPA was unilaterally inactivated in monkeys performing a free-viewing visual search for a target stimulus in an array of stimuli, impairing monkeys' ability to find the target in the array in the affected hemifield, but leaving intact their ability to make saccades to targets presented alone. Simultaneous recordings in V4 revealed that the effects of feature attention on V4 responses were eliminated or greatly reduced while leaving the effects of spatial attention on responses intact. Altogether, the results suggest that feedback from VPA modulates processing in visual cortex during attention to object features.

[1] Department of Brain and Cognitive Sciences, McGovern Institute for Brain Research, Massachusetts Institute of Technology, Cambridge, MA, USA.
*email: bichot@mit.edu

I n a cluttered visual environment, top-down control is needed to efficiently process behaviorally relevant information while filtering out irrelevant or distracting information. Such top-down processing can be based on spatial cues, e.g., knowing where a relevant object is located. However, in many instances, we do not know where a relevant object is located but know its features, and therefore need to find it based on those features. For example, if we forgot where we parked our car, we would guide our search for it based on our knowledge of its features, e.g., color, size, and shape, rather than serially examining every car on the lot to determine whether it is the right one.

Feature-based attention is known to modulate the responses of cortical visual areas such as area V4, which contains neurons selective for stimulus features such as color, orientation, and shapes[1–5], and lesions of this area lead to deficits in some aspects of visual performance and recognition[6,7]. Attending to object features biases responses in favor of stimuli that share those features in a V4 neuron's receptive field (RF)[8–13]. Using a free-viewing visual search task, we also showed that parallel (feature-based) and serial (spatial) attentional selection modulates responses in V4[14,15], consistent with most current models of visual search in which top-down feature attention biases competition in visual cortex in favor of objects sharing the target features. The feature bias occurs, in parallel, throughout the visual field, and the subject then attends serially, or foveates, likely targets until the target is found[16,17].

A key feature of many such search models is a priority map in which stimulus locations are represented in terms of their behavioral significance, e.g., similarity to the search target, or what is often termed the target template stored in memory[18–22]. Priority maps are thought to guide both spatial attention and eye movements, and they have been proposed for frontal eye fields (FEF)[23], lateral intraparietal area (LIP)[24–26], and the superior colliculus[27,28].

We previously proposed that a subregion of ventrolateral prefrontal cortex (PFC) that we termed the ventral prearcuate region, or VPA, may hold the target template, compute the match to stimuli in the visual field, and help create the priority map in FEF[29]. VPA has interconnections with IT, TEO, and possibly V4, on the one hand, and connections with FEF and other parts of PFC on the other[30,31]. We previously reported[29] that VPA cells showed selectivity for visual features, unlike FEF[23], and VPA cells have RFs ranging in size from those in FEF to those in IT cortex. The visual selectivity we observed in VPA in the context of a working memory task was consistent with other reports of object and feature selectivity in PFC in such tasks[32–35]. In our visual search task, many VPA cells responded in a stimulus-selective fashion to the cue (target) presented at the start of the trial and maintained this response for the remainder of the trial. The responses of many VPA cells were also biased in favor of stimuli in the search array that matched the target template, compared to responses to stimuli that were nontargets, and this feature bias occurred with a greater magnitude and earlier time course than in FEF, while the opposite was true for stimuli selected to be the target of an eye movement. Inactivation of VPA impaired the animals' ability to find targets, and simultaneous recordings in FEF revealed that the effects of feature attention were eliminated while the effects of spatial attention and selection in FEF were left intact. Although the results suggested that the interaction between VPA and FEF played an important role in guiding eye movements to likely targets, an important open question was whether VPA was also the direct or indirect source of top-down influences on visual cortex during feature attention[36], independent of eye movements. We addressed this question by measuring feature and spatial selection signals in area V4 during a free-viewing visual search before and after VPA inactivation.

## Results

**Behavioral performance.** Monkeys were trained to perform a free-viewing visual search task using stimuli that were conjunctions of colors and shapes as described in previous studies[14,15]. Briefly, on each trial, the animals were presented with a central cue stimulus (serving as the search target, or target template) at fixation, followed by a delay during which they held the target in memory. An array of 20 stimuli then appeared, containing both distracters and a single instance of the search target (Fig. 1a). Distractors always included two stimuli that shared the target color (same-color distractors), and two that shared the target shape (same-shape distractors), while the remaining distractors shared neither the target color nor the target shape (no-share distractors). The target color/shape conjunction stimulus and location were chosen pseudorandomly across trials. The monkeys could use free gaze to find the target in the array, and they were rewarded for maintaining fixation on the target for 800 ms continuously. Detection trials, in which a target stimulus (different than all search conjunction stimuli) was presented alone, were randomly interleaved amongst the search trials to evaluate neurons' RFs across the 20 possible stimulus locations.

We collected control and VPA inactivation data in separate sessions for both monkeys because one of the monkeys could not complete enough trials to include control and inactivation trials in the same session. We recorded neural activity in V4 during 14 control sessions in which VPA was normal (6 sessions with monkey F, and 8 with monkey J), and during 12 interleaved sessions in which a part of VPA in one hemisphere was inactivated with muscimol (6 sessions with each monkey). The region we referred to as VPA based on anatomical location likely encompasses multiple cytoarchitectonic areas such as areas 45A and 12, and even possibly area 46v. Therefore our injections at sites 2–2.5 mm apart likely did not inactivate the entirety of this region. The results from the two monkeys are presented separately, although the results were largely consistent across monkeys.

During the control sessions, monkey F found the search target on 89% of trials and monkey J on 70% of trials. They found the target after an average of 4.8 and 4.7 saccades, with an average saccadic latency of 241 and 255 ms, for monkeys F and J, respectively. A recent study from our lab showed that the attentional modulation of responses was associated with microsaccades made while animals were fixating a target in a covert spatial attention task[37]. However, similar to what has been found in other studies[38,39], these microsaccades only occurred at a median rate of 3.29 Hz, or around once every 300 ms. This was longer than the mean interval between the larger saccades made by the animals during search in the present study, thus precluding the possibility of microsaccades between the larger saccades made during search. In any case, these measures of behavioral performance indicate that the animals used the target template to efficiently guide search as they were significantly smaller than would be expected if the animals had chosen to search the 20-item display serially (i.e., finding the target in 1–20 saccades for an average of 10.5 saccades; one-sample $t$-test, $t_5 = 27.6$, $P < 10^{-5}$ and $t_7 = 42.2$, $P < 10^{-8}$, for monkeys F and J, respectively) or randomly (i.e., fixating stimuli for at least 800 ms, the required duration for reward, to determine whether they are the target; $t_5 = 78.7$, $P < 10^{-8}$ and $t_7 = 165.1$, $P < 10^{-13}$). The monkeys' gaze patterns during correct trials (Fig. 1b) also indicated that they used both the color and shape of the target to guide their search. The probability of any given stimulus being fixated during search was strongly influenced by its similarity to the target (ANOVA; $F_{3,20} = 182$ and $F_{3,28} = 1414$, $P < 10^{-13}$ and $P < 10^{-29}$, for monkeys F and J, respectively), which would otherwise be the same (i.e., 1/20) for all stimuli in the search array. Both a same-

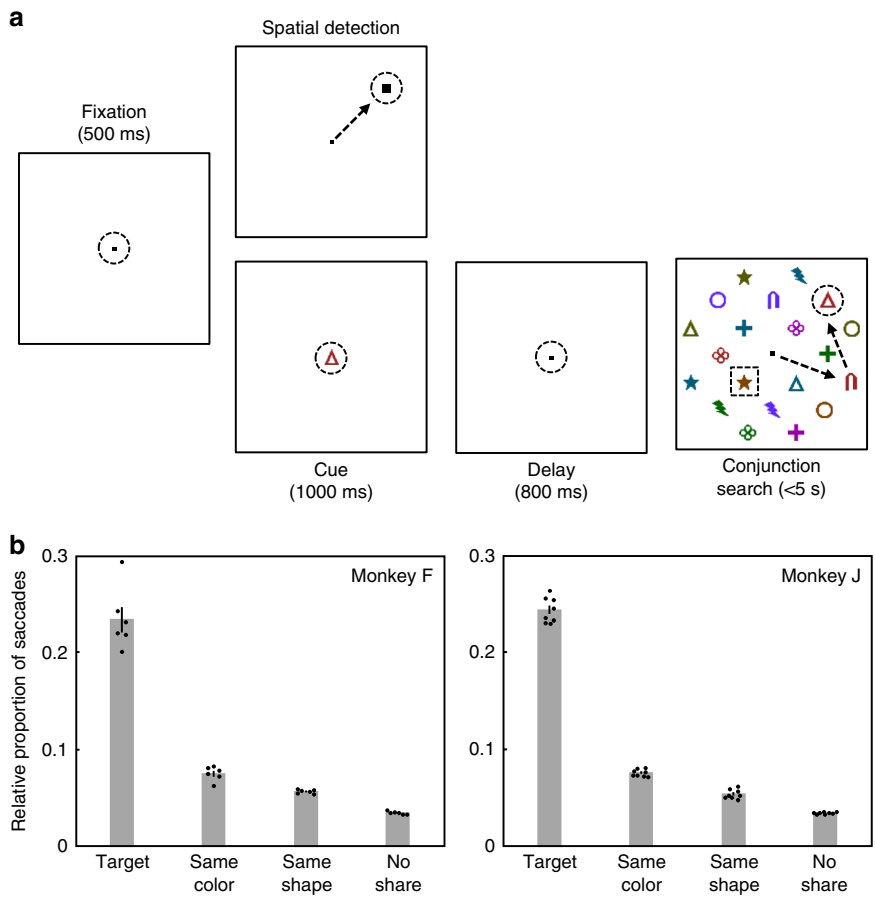

**Fig. 1 Behavioral tasks and stimulus selection during search. a** Schematic representation of behavioral tasks. Dotted circles represent the monkey's current point of fixation. Detection and search trials were pseudorandomly interleaved. For detection trials, the target was a small white square that was presented at the end of the initial fixation period. For search trials, the initial fixation period was followed by a cue period that instructed the animals which color/shape combination was the target for that trial, and following a delay period, the target (i.e., cued stimulus) was presented along with distractors. In this example of a search trial, the cue/target was the red triangle, and the animal made two saccades (represented by the sequence of black arrows) before finding the target stimulus. The dotted square represents the RF of a V4 unit. Stimuli are not drawn to scale. **b** Probability of any given stimulus being fixated as a function of its similarity to the target during correct search trials. Error bars represent ± SEM. Data points from individual sessions are shown by the black dots.

color (SMC) and a same-shape (SMS) distractor were significantly more likely to be fixated than a no-share (NS) distractor according to post hoc tests ($t$-test; SMC vs. NS: $t_{10} = 13.2$ and $t_{14} = 30.1$, $P < 10^{-6}$ and $P < 10^{-13}$, SMS vs. NS: $t_{10} = 21.9$ and $t_{14} = 11.2$, $P < 10^{-9}$ and $P < 10^{-7}$, for monkeys F and J, respectively), consistent with previous reports of stimulus selection during conjunction searches[15,40–45]. Thus, overall, on trials when the two monkeys correctly found the target, their search behavior was remarkably similar in all measures. Furthermore, all neural data analyzed in this report come from correctly performed trials.

**Effects of VPA inactivation on behavioral performance.** We first confirmed that VPA inactivation (Fig. 2) impaired the monkeys' ability to find the target in the affected hemifield (Fig. 3, Supplementary Note 1), as it had in our previous study[29]. A contralesional effect of unilateral prefrontal lesions or deactivation has been found in other studies as well[46–50]. We found a significant interaction between session type (control vs. VPA inactivation) and initial target location (ipsilateral vs. contralateral hemifield to inactivation hemisphere) for the number of saccades to find the target (mixed two-way ANOVA, interaction factor; $F_{1,10} = 28.8$ and $F_{1,12} = 8.6$, $P < 10^{-3}$ and $P = 0.012$, for monkeys F and J, respectively), the total time to find the target

($F_{1,10} = 27.0$ and $F_{1,12} = 15.3$, $P < 10^{-3}$ and $P = 0.002$), and error rates ($F_{1,10} = 27.4$ and $F_{1,12} = 46.9$, $P < 10^{-3}$ and $P < 10^{-4}$). Behavioral impairments were confirmed with post hoc tests (Supplementary Note 2). VPA inactivation also affected the general directionality of saccadic eye movements, decreasing the selection likelihood of contralateral stimuli for fixation in favor of ipsilateral stimuli (Supplementary Note 3).

In contrast to the effects of VPA inactivation on visual search, the inactivation had no significant effect on the detection trials, where the animal was rewarded for making a saccade to a single stimulus on a blank screen (Fig. 1a). We did not find a significant interaction between session type (control vs. VPA inactivation) and target location (ipsilateral vs. contralateral hemifield to inactivation hemisphere) for either the error rates (mixed two-way ANOVA, interaction factor; $F_{1,10} = 0.01$ and $F_{1,12} = 0.01$, $P = 0.92$ for both monkeys), or saccade latencies ($F_{1,10} = 0.06$ and $F_{1,12} = 0.1$, $P = 0.81$ and 0.73, for monkeys F and J, respectively).

**Effects of VPA inactivation on feature-based selection.** We next asked whether unilateral VPA inactivation affected feature-based selection in V4. To separate out the effects of feature-based and spatial-based attention on neural responses, we used a strategy that has been used in previous studies of FEF, VPA, and V4[14,15,29,41,51]. To isolate the effects of feature attention, we

examined responses to the stimulus in the RF at times during the trial when the animal was preparing a saccade to a stimulus outside the RF. With spatial attention directed outside the RF, we asked whether the response to the RF stimulus varied according to whether the RF stimulus matched the target template (red lines in Fig. 4) or did not share any features with the target (i.e., a no-share distractor; blue lines in Fig. 4). For example, we asked

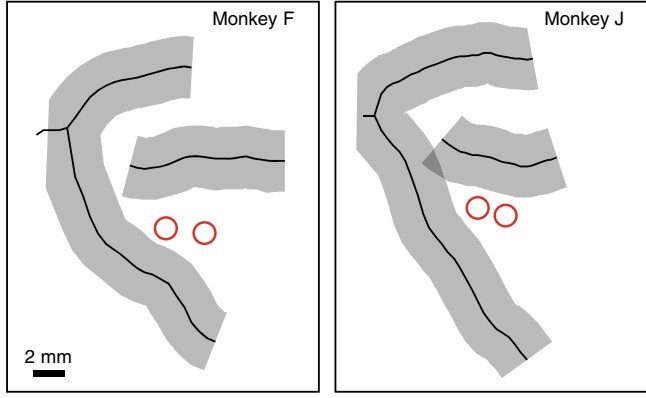

**Fig. 2 Anatomical reconstruction of muscimol injection sites for each monkey.** Injection sites are shown by the red circles. The reconstruction was based on coronal sections from structural MRI images (3 T MPRAGE, 500 µm isotropic).

whether the response to a red circle inside the RF was different when the animal was searching for a red square compared to when it was searching for a green triangle. If the response to a stimulus in the RF varied according to what the animal was searching for, it would be evidence for feature attention. This approach to measuring feature-based attentional effects avoids potential confounds between feature and spatial attention found in previous studies of attentional effects during visual search or traditional cueing paradigms (for a review, see ref. [52]).

Neural responses to a stimulus appearing in the RF with the onset of the search array were typically larger than those elicited by a stimulus entering the RF as a result of a saccade. We therefore separately considered responses during two activity periods: (1) from the onset of the array until the first saccade following array onset, and (2) from the end of any saccade to the start of the next one (e.g., end of first saccade to start of second saccade and end of second saccade to start of third saccade). For each monkey, responses were calculated for the duration of the average latency of the first saccade across all sessions (230 ms for monkey F, and 304 ms for monkey J) rounded to the nearest multiple of 50 ms (i.e., 250 ms for monkey F and 300 ms for monkey J). Only activity from correct trials were used in the analysis of feature-based selection effects.

We recorded the activity of 127 V4 units (53 in monkey F and 74 in monkey J) during VPA inactivation sessions, and compared it to the activity of 135 units (52 in monkey F and 83 in monkey J) recorded during control sessions. The average RF eccentricities

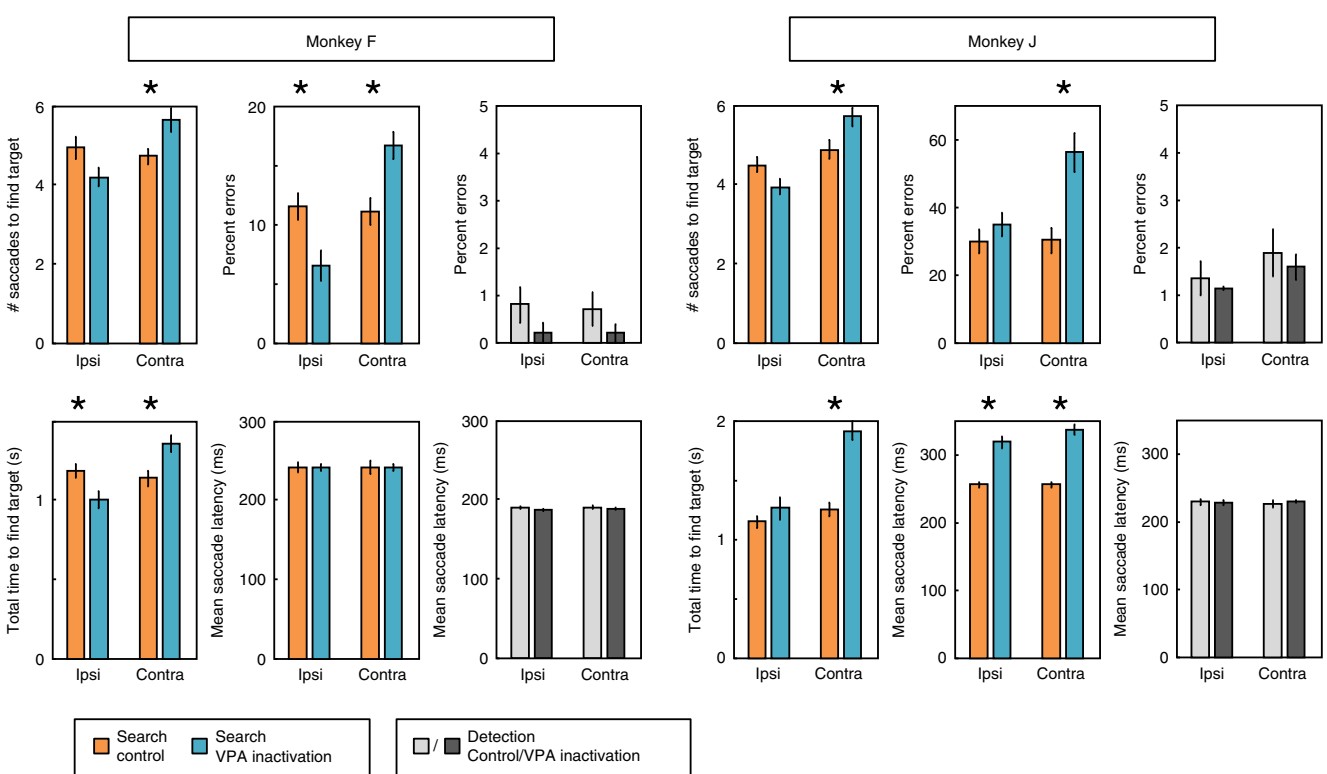

**Fig. 3 Effects of VPA inactivation on behavioral performance.** Effects of VPA inactivation on behavioral performance during search trials are shown with colored bars for the two monkeys separately. Across session averages of the number of saccades to find the target, the total time to find the target, the error rate, and the saccade latency during control sessions (orange bars) and sessions in which VPA was inactivated (blue bars) are plotted as a function of target location relative to the hemisphere in which VPA was inactivated. The averages of the number of saccades to find the target, the total time to find the target, and the saccade latency were calculated using correct trials only. Effects of VPA inactivation on behavioral performance during detection trials are shown with grayscale bars for the two monkeys separately. Across session averages of the error rate and the saccade latency during control sessions (light gray bars) and sessions in which VPA was inactivated (dark gray bars) are plotted as a function of target location relative to the hemisphere in which VPA was inactivated. Error bars represent ± SEM, and asterisks (*) mark significant effects of inactivation based on t-tests (i.e., $P < 0.05$). Data points from individual sessions are shown in Supplementary Fig. 1.

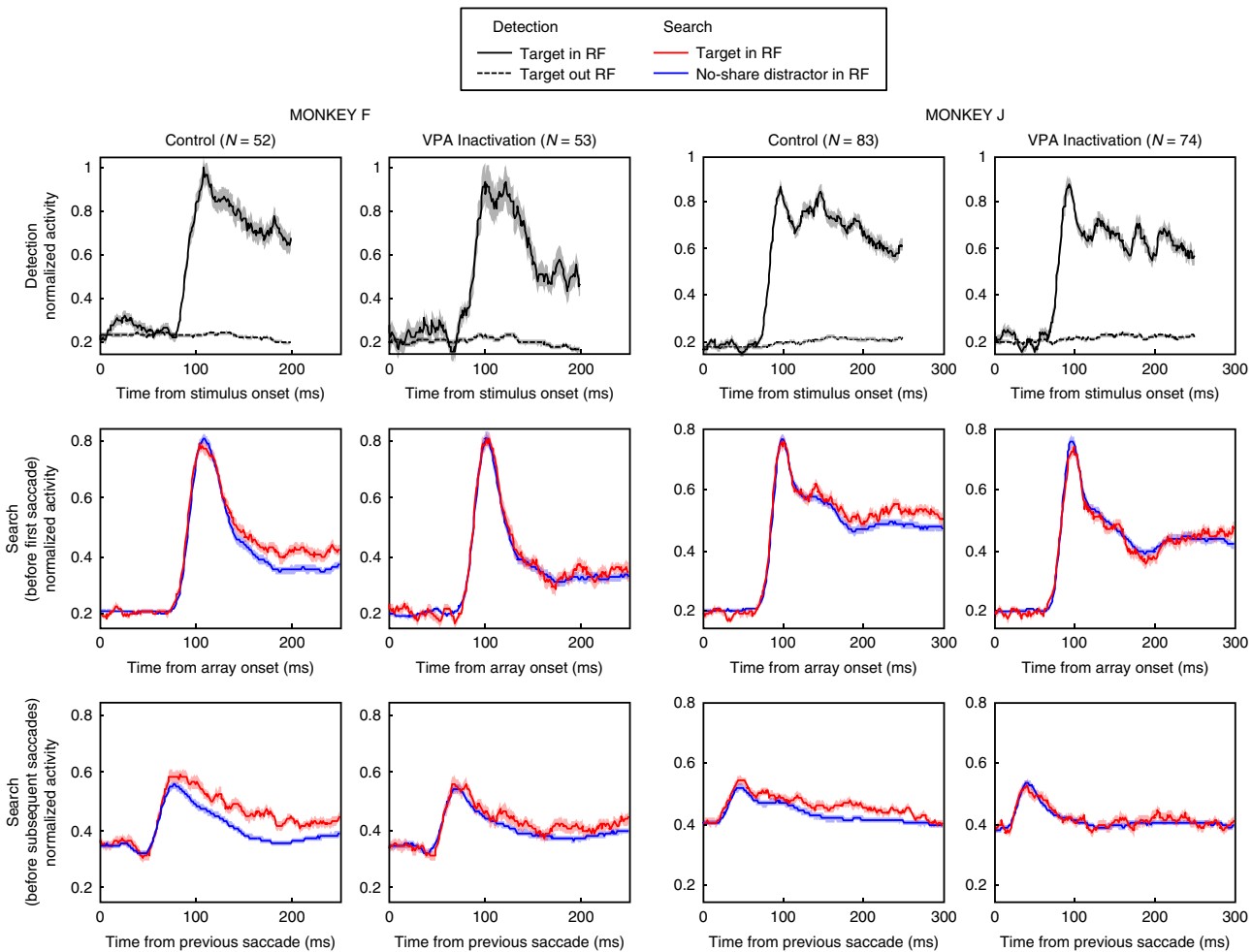

**Fig. 4 Effects of VPA inactivation on feature selection in area V4.** Normalized firing rates averaged across the population of recorded V4 neurons are shown for detection trials (top row) and search trials (middle and bottom rows) for the two monkeys separately. For each monkey, the data from the control sessions are shown on the left, and the data from sessions in which VPA was inactivated are shown on the right, with the number of units contributing to each type of session represented by N. For detection trials, activity is plotted when the target was in a V4 unit's RF (solid black line) and when the target was outside the unit's RF (dotted black line). For search trials, activity is plotted when the target was in the RF but the saccade was made to a distractor outside the RF (red lines), and when the target was outside the RF (and a no-share distractor was in the RF) and the saccade was made to a distractor outside the RF (blue lines). Because activity immediately following array onset contains a strong evoked response to the onset, we plotted activity following the array onset and leading up to the first saccade separately from activity following subsequent fixations/saccades. Activity in all conditions shown (including detection trials) for a given neuron were normalized by the maximum response elicited by that neuron after array onset and before the first saccade (i.e., activity shown in the middle row). Only spikes occurring prior to saccade initiation were used in the analyses. SEM (±) at each time point is indicated by shading over the lines.

during control sessions were 4.4 and 4.6 dva for monkeys F and J, respectively, and the average RF eccentricities during inactivation sessions were 4.5 and 4.8 dva for the same monkeys. The RF locations across sessions were similar, and there was no significant difference in average RF eccentricities between control and inactivation sessions for either monkey ($t$-test; $t_{10} = 0.3$ and $t_{12} = 0.7$, $P = 0.79$ and $P = 0.50$, for monkeys F and J, respectively). In both control and VPA inactivation sessions, units responded strongly when a detection target was in neurons' RF and very weakly if at all when it was outside neurons' RF for both monkeys, as shown by the population response functions in the top row of Fig. 4. We analyzed average activity starting from 70 ms (to encompass the onset of visual responses) to the end of time for which activity is plotted (200 and 250 ms for monkeys F and J, respectively, after rounding average saccade latencies during detection trials for each monkey to the nearest multiple of 50 ms), and found that VPA inactivation did not affect V4

responses during detection trials ($t$ test; $t_{103} = 1.6$ and $t_{155} = 1.3$, $P = 0.10$ and 0.19 for monkeys F and J, respectively).

We next considered the effects of feature attention on the search trials, immediately following the array onset (Fig. 4, second row). In the control sessions, the initial transient population response to targets and no-share distracters was the same, but beginning about 130 ms after stimulus onset, the response to target stimuli grew larger than to the same stimuli when they were no-share distracters. This effect of feature attention on V4 responses was consistent with that found in a previous study of V4 using a similar task[15]. However, in the VPA inactivation sessions, the population response to targets and the same stimuli as no-share distracters remained the same until the first saccade. To quantify the feature selection effect, we averaged responses from 150 ms to the end of the analysis window. Activity is plotted and shown for each monkey (250 ms and 300 ms for monkey F and J, respectively). Average responses in this interval for

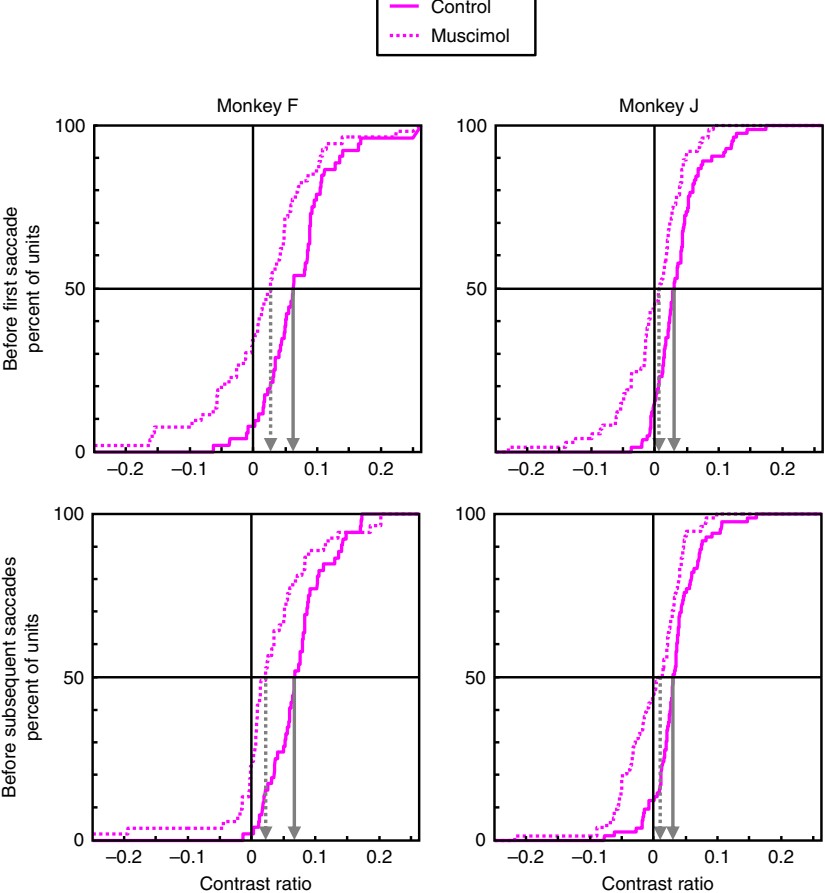

**Fig. 5 Effects of VPA inactivation on feature selection contrast ratios in area V4.** Cumulative distributions of attentional contrast ratios for control (solid lines) and VPA inactivation (dotted lines) sessions. Contrast ratios were computed for each neuron as the difference in responses between the two attentional conditions (i.e., target in RF vs. distractor in RF), divided by the sum of those responses. The first point represents all contrast ratios of less than −0.25, and the last point represents all contrast ratios greater than 0.25. The median contrast values for each session type is shown by the arrows (solid for control and dotted for inactivation sessions).

individual neurons are shown in Supplementary Fig. 2. We found a strong effect of VPA inactivation on feature selection (mixed two-way ANOVA; interaction of session type and stimulus type in RF, $F_{1,103} = 24.2$ and $F_{1,155} = 32.1$, $P < 10^{-5}$ and $P < 10^{-7}$ for monkeys F and J, respectively). Follow-up post hoc tests showed that while there were strong feature selection effects during control sessions (t test; $t_{51} = 8.9$ and $t_{82} = 8.8$, $P < 10^{-11}$ and $P < 10^{-12}$ for monkeys F and J, respectively), feature selection effects in V4 were eliminated when VPA was inactivated (t-test; $t_{52} = 1.7$ and $t_{73} = 0.02$, $P = 0.1$ and $P = 0.98$). Thus, the effects of feature attention on V4 responses to array onset appeared to be eliminated following the inactivation.

Similar results were found for neuronal responses following the first saccade and subsequent saccades during the trial (Fig. 4, bottom row). Feature attention enhanced responses to RF stimuli in the control sessions, but these effects were largely eliminated following VPA inactivation. To quantify these effects, we averaged responses from 100 ms (responses were modulated by attention earlier than found for the first saccade) to the end of the analysis. Average responses in this interval for individual neurons are shown in Supplementary Fig. 2. Again, we found a strong effect of VPA inactivation on feature selection (mixed two-way ANOVA; interaction of session type and stimulus type in RF, $F_{1,103} = 12.6$ and $F_{1,155} = 20.3$, $P < 10^{-3}$ and $P < 10^{-4}$ for monkeys F and J, respectively). Follow-up post hoc tests showed that while there were strong feature selection effects during

control sessions (t-test; $t_{51} = 10.8$ and $t_{82} = 8.6$, $P < 10^{-14}$ and $P < 10^{-12}$ for monkeys F and J, respectively), feature selection effects in V4 when VPA was inactivated were either eliminated (monkey J, t-test; $t_{73} = 0.9$, $P = 0.36$) or significantly reduced (as shown by the significant interaction effect) for monkey F (t-test; $t_{52} = 3.4$, $P = 0.001$). The effects of VPA inactivation on feature-based selection in V4 before the first saccade or subsequent saccades were not dependent on the selectivity of V4 neurons for the colors and shapes used in the experiment (Supplementary Note 4).

We confirmed these effects using the contrast ratio measure of feature-based attentional modulation and comparing median values non-parametrically. Contrast ratios were computed as the difference between responses in the attentional conditions divided by the sum of those responses (i.e., response to target in RF vs. response to no-share distractor in RF as calculated in the analyses above). Cumulative distributions of contrast ratios for each monkey separated by period before first vs. subsequent saccades (as in the analyses above) are shown in Fig. 5 (the non-cumulative bar distributions are shown in Supplementary Fig. 2). We found a significant decrease of attentional contrast ratios during inactivation sessions for both monkeys before the first saccade (monkey F, median contrast ratio: control = 0.063 vs. inactivation = 0.027, Wilcoxon rank sum test: $P < 10^{-3}$; monkey J, control = 0.030 vs. inactivation = 0.007, $P < 10^{-5}$) as well as subsequent saccades (monkey F, control = 0.068 vs. inactivation = 0.022, $P < 10^{-4}$;

monkey J, control = 0.030 vs. inactivation = 0.010, $P < 10^{-4}$). Furthermore, the median contrast ratio during inactivation sessions was significantly different than zero only for monkey F and only for the period before subsequent saccades (Wilcoxon signed rank test, $P < 10^{-4}$). The median contrast ratio during control sessions was significantly different than zero for both time periods and monkeys (all $Ps < 10^{-8}$).

As shown in Fig. 3, saccade latencies increased for monkey J during VPA inactivation sessions. To address the possibility that feature selection occurred later in trials with longer latencies, which might have resulted in underestimating the magnitude of effects during inactivation sessions compared to control session due to using fixed time windows aligned on array presentation, we repeated the analyses of feature selection with activity aligned on the onset of saccades for both monkeys (Supplementary Fig. 3). To quantify feature selection effects, we measured average neural activity for each neuron in a 100 ms time window before the first saccade and a 150 ms before subsequent saccades. Average responses in these intervals for individual neurons are shown in Supplementary Fig. 4. Results were unchanged with analyses conducted with activity aligned on the time of saccade onset. We found a strong effect of VPA inactivation on feature selection for both the first saccade (mixed two-way ANOVA; interaction of session type and stimulus type in RF, $F_{1,103} = 8.9$ and $F_{1,155} = 15.1$, $P = 0.003$ and $P = 10^{-4}$ for monkeys F and J, respectively) and subsequent saccades ($F_{1,103} = 6.4$ and $F_{1,155} = 10.1$, $P = 0.013$ and $P = 0.002$). Furthermore, the significant feature selection found during control sessions for both monkeys before the onset of the first saccade (t-test; $t_{51} = 5.9$ and $t_{82} = 8.3$, $P < 10^{-6}$ and $P < 10^{-11}$ for monkeys F and J, respectively) was eliminated during inactivation sessions ($t_{52} = 1.0$ and $t_{73} = 0.3$, $P = 0.33$ and $P = 0.76$). On subsequent saccades, the effects of feature selection on neuronal responses was eliminated after inactivation for monkey J (t-test; control: $t_{82} = 9.1$, $P < 10^{-13}$; inactivation: $t_{73} = 1.1$, $P = 0.27$), and significantly reduced for monkey F (again, as shown by the significant interaction effect; control: $t_{51} = 7.8$, $P < 10^{-9}$; inactivation: $t_{52} = 2.3$, $P = 0.024$).

As described above, during detection trials, overall visual responses during an analysis interval starting ~70 ms after stimulus onset were not affected by inactivation for either monkey. We also analyzed visual responses during detection trials using the later analysis interval that was used on the search trials, i.e., beginning at 150 ms after array onset. We again found no significant effect of inactivation on responses in the detection trials in monkey J (t test; $t_{155} = 1.5$, $P = 0.15$), but there was a significantly smaller response to the RF stimulus in monkey F ($t_{103} = 3.7$, $P < 0.001$). However, for this monkey, there was no correlation between detection trial responses in this later time interval and the magnitude of feature attention effects in either session type (control: correlation coefficient $R = -0.11$, $P = 0.45$; inactivation: $R = 0.09$, $P = 0.50$). The absence of a significant correlation strongly suggests that the effects of VPA inactivation on responses during feature attention were not caused by overall lower visual responsiveness; however, to rule out a potential confound more directly, we iteratively removed cells with the highest responses in detection trials in control sessions and cells with the lowest responses during detection trials in the inactivation sessions for this monkey until there was no significant difference in overall visual responsiveness during detection trials between the remaining populations of cells (6 cells removed from each population; t test; $t_{91} = 1.8$, $P = 0.07$). Results were unchanged with the populations matched for visual responsiveness during detection. For this same subsampled population, we found a strong effect of VPA inactivation on feature selection (mixed two-way ANOVA; interaction of session

type and stimulus type in RF, $F_{1,91} = 24.8$, $P < 10^{-5}$), and follow-up post hoc tests showed that while there were strong feature selection effects during control sessions (t-test; $t_{45} = 8.7$, $P < 10^{-10}$), feature selection effects in V4 were eliminated when VPA was inactivated (t-test; $t_{46} = 1.8$, $P = 0.08$). Thus, the effects of feature attention on V4 responses to array onset appeared to be eliminated following the inactivation, regardless of magnitude of visual responses during detection trials.

Finally, we analyzed the effects of inactivation on color (Supplementary Fig. 5) and shape (Supplementary Fig. 6) feature selection by comparing the neurons' response to same-color and same-shape distractors in their RF, respectively, to their response to no-share distractors in their RF (all responses when the saccade was made away from the RF). Overall, the enhancement for distractors sharing a target feature was much smaller than for the target. In fact, we found no significant enhancement for same-shape distractors in either control or inactivation sessions, with no interaction between session types for either monkey. The same was true when we compared contrast ratios. In contrast, there was a modest and significant enhancement for same-color distractors during control sessions, and this effect was eliminated during inactivation sessions for both monkeys; the results were confirmed using contrast ratios. Thus, much like the object feature modulation observed for the target in the RF, the color feature modulation observed for a distractor sharing the target color was eliminated after inactivation of VPA.

**Effects of VPA inactivation on spatial selection.** To evaluate the effects of spatial attention or targeting for saccades, we compared responses when the animal made a saccade to the RF stimulus (green lines in Fig. 6) to responses when the animal made a saccade to a stimulus outside the RF (blue lines in Fig. 6). We did not have enough trials (especially during the inactivation sessions when animals made fewer saccades towards the RF in the affected hemifield) to match the features of the stimulus in the RF on a cell-by-cell basis when animals made a saccade to it or away from it; however, across sessions there was no difference in the RF stimulus features (i.e., color or shape) relative to saccade direction or session type (t-test with Bonferroni correction, all $Ps > 0.05$). Also, to completely dissociate spatial selection from feature-based selection, we did not include fixations when the target stimulus was in the RF. Furthermore, there was no significant difference in the distance from saccade location to the target's location between control and inactivation sessions whether the animals made a saccade to the RF or away from the RF (t-test; monkey F—saccade to RF: $t_{10} = 0.28$, $P = 0.79$, saccade away from RF, $t_{10} = 0.37$, $P = 0.72$; monkey J: saccade to RF, $t_{12} = 0.25$, $P = 0.80$, saccade away from RF, $t_{12} = 0.44$, $P = 0.67$).

We again analyzed the neural activity for the first fixation after array onset and before the first saccade separately from activity for subsequent fixations and saccades. To quantify the effects of spatial selection, we averaged activity in the same time windows that were used for feature selection above (i.e., 150–250 ms and 100–250 ms for monkey F, and 150–300 ms and 100–300 ms for monkey J, for activity after array onset and activity after subsequent fixations, respectively). Average responses in these intervals for individual neurons are shown in Supplementary Fig. 7. We found no significant effects of VPA inactivation on spatial selection following array onset (mixed two-way ANOVA; interaction of session type and saccade to/away from RF stimulus, $F_{1,98} = 1.1$ and $F_{1,155} = 2.5$, $P = 0.29$ and $P = 0.12$ for monkeys F and J, respectively), with significant spatial selection effects in both control (post hoc t-test; $t_{51} = 5.7$ and $t_{82} = 8.1$, $P < 10^{-6}$ and $P < 10^{-11}$, for monkeys F and J, respectively) and inactivation ($t_{47} = 3.6$ and $t_{73} = 6.8$, $P < 10^{-3}$ and $P < 10^{-8}$) sessions. The

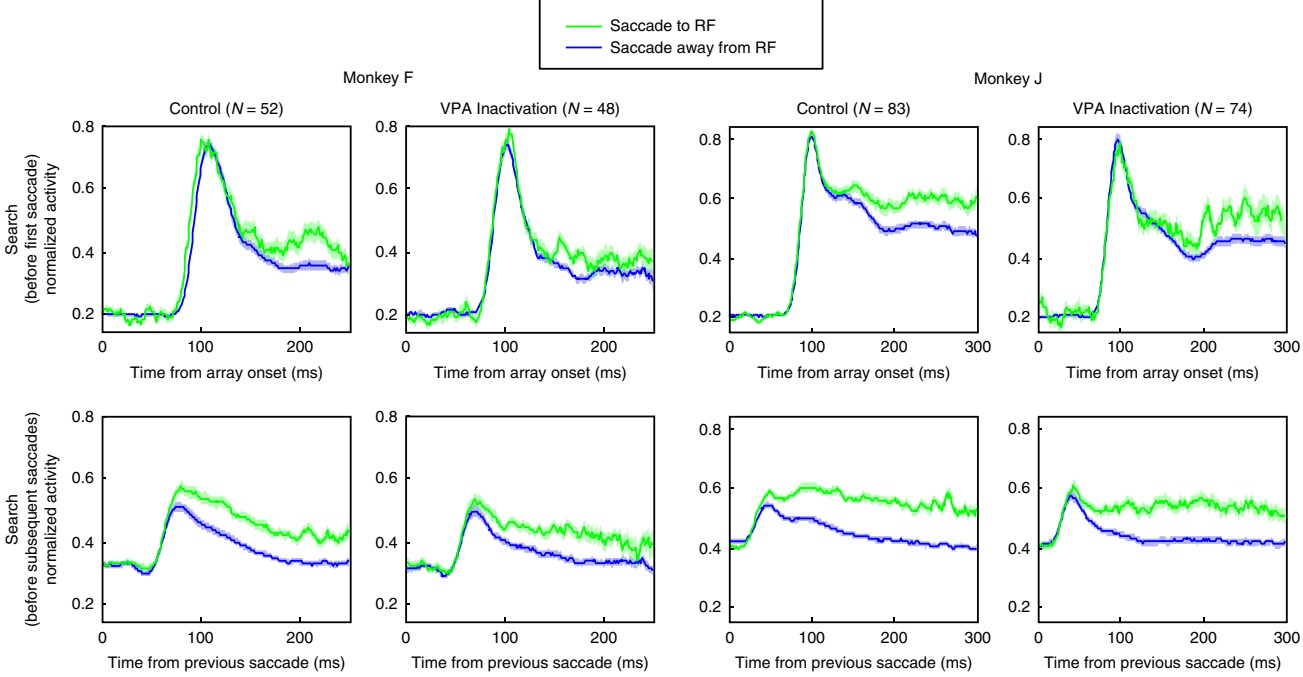

**Fig. 6 Effects of VPA inactivation on spatial selection in area V4.** Activity is plotted when the saccade was made to a stimulus in the RF (green lines), and when the saccade was made to a stimulus outside the RF (blue lines). Five neurons from VPA inactivation sessions with monkey F did not contribute enough trials (i.e., minimum of five trials) in which the animal made a saccade to the RF stimulus on the first saccade and were not included in the analyses. All other conventions as in Fig. 4.

same was true for subsequent fixations, with no significant effects of VPA inactivation on spatial selection (mixed two-way ANOVA; interaction of session type and saccade to/away from RF stimulus, $F_{1,98} = 2.1$ and $F_{1,155} = 1.0$, $P = 0.16$ and $P = 0.32$ for monkeys F and J, respectively), and significant spatial selection effects in both control (post hoc $t$-test; $t_{51} = 12.2$ and $t_{82} = 16.8$, $P < 10^{-15}$ and $P < 10^{-27}$, for monkeys F and J, respectively) and inactivation ($t_{47} = 7.9$ and $t_{73} = 14.2$, $P < 10^{-9}$ and $P < 10^{-21}$) sessions. Thus, VPA inactivation appeared to eliminate the effects of feature attention in V4 while leaving the effects of spatial attention intact.

Again, we confirmed these results using the contrast ratio measure of spatial attentional modulation. We compared median values using a nonparametric rank sum test. Contrast ratios were computed as the difference between responses in the attentional conditions divided by the sum of those responses (i.e., saccade to RF vs. saccade away from RF as calculated in the analyses above). Cumulative distributions of contrast ratios for each monkey separated by period before first vs. subsequent saccades (as in the analyses above) are shown in Fig. 7 (the noncumulative bar distributions are shown in Supplementary Fig. 7). We found no significant difference in median attentional contrast ratios between inactivation and control sessions in either monkey before the first saccade (monkey F, median contrast ratio: control = 0.069 vs. inactivation = 0.047, Wilcoxon rank sum test: $P = 0.24$; monkey J, control = 0.059 vs. inactivation = 0.061, $P = 0.89$) or subsequent saccades (monkey F, control = 0.092 vs. inactivation = 0.094, $P = 0.79$; monkey J, control = 0.119 vs. inactivation = 0.109, $P = 0.23$). Furthermore, the median contrast ratio was significantly different than zero for both monkeys in both session types and saccade analysis period (Wilcoxon signed rank test, all $Ps < 10^{-3}$).

## Discussion

We tested whether VPA is a source of top-down feedback to V4 for feature attention during search. Inactivation of VPA not only impaired the animals' ability to find the target in the contralateral visual field, but also eliminated or greatly reduced feature-based attentional selection in area V4 in the same hemisphere. The fact that the impairments following VPA inactivation were confined to the contralateral hemifield with no change or even improvement in the ipsilateral field, and the fact that the animals did not know beforehand in which hemifield the target would be located, argue against the possibility that the neural effects were the result of a general reduction in effort. While our observed contralesional deficits may at first appear at odds with studies (including one of our own) showing that feature attention acts in parallel across the entire visual field[14,53–55], this seeming discrepancy is due to the fact that we only unilaterally inactivated VPA. Our finding of contralesional deficits following unilateral VPA deactivation in this study are consistent with previous studies showing lateralized deficits following unilateral PFC lesions in macaques[46,47]. Like-wise, lateralized deficits in attention and working memory have been found following unilateral lesions in humans[49,50]. VPA in each hemisphere apparently provides feature bias in the contralateral hemifield. The fact that the effects of feature attention are found globally, across both hemifields, is presumably due to a coordination of activity related to the target template across VPA in the two hemispheres. Had we bilaterally inactivated VPA, it seems likely that we would have observed widespread deficits covering the entire visual field. Altogether, our results show that VPA is critical for the effects of feature attention in V4, although we cannot say whether the influence on V4 is direct or mediated by other brain structures.

While VPA has strong connections with IT cortex and FEF, it is unclear if it has direct connections with V4[30,31,56]. It is possible that VPA influences V4 indirectly, through direct influences on IT cortex, which are then fed back to V4. An indirect influence through IT cortex might explain why the effects of feature attention are generally expressed globally across the visual field[14,54,55,57–59], consistent with the large RFs of IT neurons. We previously found that the effects of feature attention during

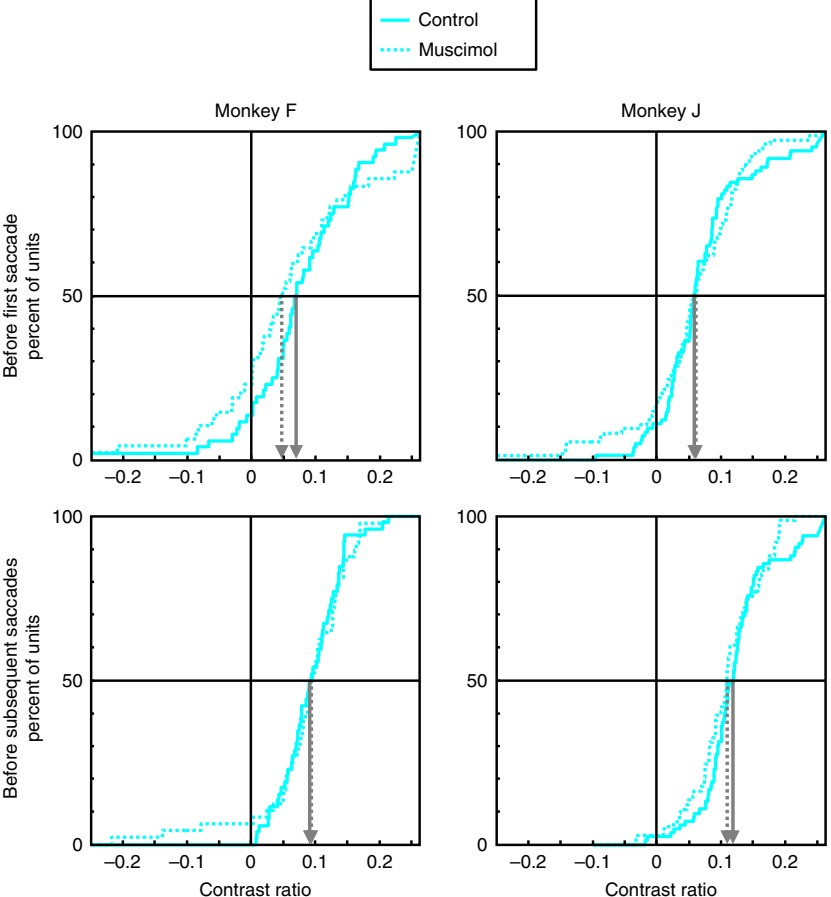

**Fig. 7 Effects of VPA inactivation on spatial selection contrast ratios in area V4.** Cumulative distributions of attentional contrast ratios for control (solid lines) and VPA inactivation (dotted lines) sessions. Contrast ratios were computed for each neuron as the difference in responses between the two attentional conditions (i.e., saccade to stimulus in RF vs. saccade to stimulus outside RF), divided by the sum of those responses. The first point represents all contrast ratios of less than −0.25, and the last point represents all contrast ratios greater than 0.25. The median contrast values for each session type is shown by the arrows (solid for control and dotted for inactivation sessions).

search were weak in anterior IT cortex[29], but we did not record from posterior IT cortex, or TEO.

An alternative possibility is suggested by the fact that V4 has robust reciprocal connections with FEF and LIP[56,60], regions that, as mentioned earlier, are thought to represent salience/priority maps for guiding spatial attention and eye movements to targets derived from both bottom-up and top-down factors (for review, see ref. [61]). It is conceivable that VPA exerts its influence on feature-based selection in retinotopic areas of visual cortex indirectly through its influences on priority maps, such as in FEF or LIP[62,63]. Spatially directed feedback from areas such as FEF could enhance processing in visual areas at all locations containing features shared with the target. Indeed, in a series of studies, Moore and colleagues have shown that the response properties of V4 neurons area affected by microstimulation or inactivation of FEF[64–67]. While this mechanism might be effective in visual search, where every stimulus has a unique location, a spatial-based mechanism could not, however, easily explain the effects of feature attention in stimulus configurations where feature-defined target and distracters are not readily be separated by spatial location, e.g., transparent displays or stimuli consisting of overlapping dot fields. Resolving these questions will require manipulation of anatomical pathways as well as testing different stimulus configurations.

Regardless of the pathway by which VPA influences V4, our results provide strong evidence for a prefrontal mechanism for feature-based attention. Through its influence on FEF, VPA could influence the choice of likely targets to fixate during visual search and through its influence on V4 (and presumably other cortical areas), it could also influence the perception of stimuli during search.

## Methods

**Subjects and surgical procedures.** Two adult male rhesus monkeys weighing 8–13 kg were used. The animals were cared for in accordance with the National Institutes of Health Guide for the Care and Use of Laboratory Animals and the guidelines of the MIT Animal Care and Use Committee. All surgical procedures were carried out under anesthesia, and animals received antibiotics and analgesics after surgery. Under aseptic conditions, monkeys were implanted with a headpost and chambers that allowed access to brain regions for neural recording and inactivation.

**Behavioral tasks.** The experiments were under the control of a PC computer using MonkeyLogic software (University of Chicago, IL), which presented the stimuli, monitored eye movements, and triggered the delivery of the reward. Monkeys were seated in an enclosed chair and their eye position was monitored using an EyeLink II (SR Research Ltd., ON, Canada) infrared, video-based system for monkey F and an ISCAN ETL-200 (ISCAN, Inc., Woburn, MA) infrared, video-based system for monkey J. Stimuli were presented on an LCD video monitor (120 Hz, 1680 × 1050 resolution) viewed binocularly at a distance of 57 cm in a dark isolation box (Crist Instrument Co., MD).

Each recording session started with a RF mapping task. This task was used to determine the RF centers of units on the different recording contacts in order to optimize the array locations of stimuli in the search task such that the majority of units responded to those stimuli. A 20 × 20 dva extent of the visual field around

fixation (i.e., ±10 dva horizontally and ± vertically) divided into 2 × 2 or 1 × 1 dva grid of non-overlapping locations was mapped using a white square matching the grid size. During each mapping trial, the white square was sequentially flashed at 16 locations chosen pseudo-randomly while the animals maintained fixation on a small central fixation spot. The mapping stimulus remained on for 100 ms at each location, and there was a 100 ms delay before presentation at the next location. A block of mapping trials was complete when ten stimulus presentations were obtained at all grid locations.

Stimuli for the free-viewing search task were conjunctions of seven colors and seven shapes. The colors and shapes were fixed and remained the same throughout the study. The colors of the stimuli were matched for luminance (24 cd/m$^2$), and their shapes were matched for the number of pixels different from the gray background (12 cd/m$^2$), subtending an area of approximately 1.5 × 1.5 dva. For each recording session, the stimuli were positioned at 20 fixed locations selected based on the RF locations of the recorded neurons such that one stimulus typically fell in the center of neurons' RF when gaze landed at most stimulus locations. After fixating a small, white central fixation point for 800 ms, the monkeys were presented with a central cue that informed it of the stimulus selected as the search target for that trial. The cue stimulus stayed on for 1000 ms, after which time it was extinguished and replaced by the fixation spot for another 800 ms. The monkeys were required to hold fixation at the center of the screen during this delay period. At the end of the delay period, the fixation spot was extinguished and, simultaneously, the target was presented among distractors. The monkeys were required to fixate the target stimulus for 800 ms continuously to receive a reward. The animals had 5 s from search array onset to find the target, and no constraints were placed on their search behavior in order to allow them to conduct the search naturally (e.g., they could fixate distractors as long as they wanted within the trial). In other words, a search trial was considered an error only if an animal never fixated the search target continuously for 800 ms within the 5 s search duration. The intertrial interval was 1 s. The distractors always included two that shared the target color (i.e., same-color distractors), two that shared the target shape (i.e., same-shape distractors), and the remaining 15 that shared neither the target color nor the target shape (i.e., no-share distractors). Furthermore, the distractor colors and shapes were selected such that only 2 or 3 stimuli of each color or shape were present in the search array. Once the target location was selected for a given trial, distractors were assigned randomly to the remaining locations. An experimental block of search trials consisted of an animal successfully finding each target color/shape combination at each location once (i.e., 7 colors × 7 shapes × 20 positions = 980 correct trials). The target location and identity were chosen pseudorandomly from trial to trial such that all desired target identity and target location combinations were completed successfully.

Detection trials were pseudo-randomly interleaved with the search trials such that 10 successful detection trials were completed at each location within an experimental block of trials (i.e., 980 search trials + 200 detection trials = 1180 correct trials). For detection trials, there was no cue or delay period, and the detection target was presented at the end of the initial fixation period. The detection target was a gray filled square that matched the search stimuli in color luminance and the number of pixels different from background. From the time the animals' gaze left the fixation window, they had 50 ms to enter the target window and keep fixation at the target location until reward (i.e., multiple saccades were not allowed during detection trials in order to accurately map the properties of neurons' RF).

**Neural recordings and inactivation.** Recordings began only after the monkeys were fully proficient in the search task and performance (i.e., accuracy, reaction times, and number of saccades to find the target) was stable from session to session. Only sessions in which monkeys correctly completed at least half of the trials of an experimental block (i.e., 1180/2 = 590) were used in the analyses. Recordings were conducted with a multi-contact laminar electrodes (Plexon Inc., Dallas, TX) with 16 contacts spaced at 150 μm intervals. The electrodes were advanced manually using custom-made screw mini-microdrives mounted on a plastic grid, similar to the ones used by Miller and colleagues[68]. Neural signals were amplified, band-pass filtered, and digitized using the Omniplex system (Plexon Inc.). Neural data was sorted offline using the Offline Sorter software (Plexon Inc.). Due to the long duration of sessions, it was difficult to keep isolation on a single neuron; thus, the majority of the data are multi-unit activity and are presented as such. The possibility of overlapping neural activity from adjacent contacts using this recording method has been addressed and shown to not occur significantly in a previous report[29].

A grid system with holes 1 mm apart was used inside all the recording or inactivation chambers to guide electrode penetrations and localize them relative to structural MRI images (3 T MPRAGE, 500 μm isotropic). Ventral pre-arcuate (VPA) inactivation sites were on the pre-arcuate gyrus, approximately 2.5–4 mm anterior to the arcuate sulcus and ventral to the principal sulcus, and the penetrations did not enter either the arcuate sulcus or the principal sulcus (i.e., white matter was reached by the expected depth). The two inactivation locations in monkey F were the same as the ones in our previous study[29]. V4 recording locations were in the lunate gyrus.

Muscimol (5 μg/μl) was injected in VPA in inactivation sessions. In such a session, we made injections of 1 μl at three different depths and two locations within VPA. The injections started at the deep layers, and subsequent injections

were made by retracting the cannulas by steps of 700 μm. The injections were made at a rate of 0.05 μl/min with a 5-min wait between injections, and data collection began 35 min after the last injection. No experimental (i.e., recording) session took place the day after a inactivation session in order to ensure that brain activity returned to normal before another recording session.

**Data analysis.** Data were analyzed using MATLAB (MathWorks, Natick, MA). Spike density functions were generated by computing, at each time point, the average number of spikes in a 10-ms window around that time point (i.e., a 10-ms smoothing of average spike counts at each time point). We used a mixed (between- and within-subjects factors) two-way ANOVA to test the statistical significance of the effects of VPA inactivation. For behavioral effects, the between-sessions factor was the session type (i.e., control vs. VPA inactivation), and the within-sessions (repeated measures) factor was the visual hemifield in which the target was located relative to the hemisphere in which VPA was inactivated (i.e., ipsilateral vs. contralateral). For effects on feature selection in V4, the between-neurons factor was control vs. VPA inactivation session, and the within-neurons factor was activity when the target vs. a no-share distractor was in the neurons' RF. For effects on spatial selection in V4, the between-neurons factor was control vs. VPA inactivation session, and the within-neurons factor was activity when a saccade was made to a stimulus in the neurons' RF vs. a saccade to a stimulus outside the neurons' RF. The significance of the interaction factor (i.e., between × within factors) was used to determine whether inactivation of VPA, compared to control, affected how well monkeys found the target in the contralateral vs. ipsilateral hemifields, and how much modulation related to feature or spatial selection was present in V4. All $t$-tests were two-tailed.

**Reporting summary.** Further information on research design is available in the Nature Research Reporting Summary linked to this article.

## Data availability
The data that support the findings of this study are available from the corresponding author upon reasonable request.

## Code availability
The code that support the findings of this study is available from the corresponding author upon reasonable request.

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

## Acknowledgements

This work was funded by U.S. National Eye Institute grant EY017921 and NSF grant CCF 1317348 (to R.D.).

## Author contributions

N.P.B. and R.D. designed the research. N.P.B., R.X., A.G., and M.L.W. performed the experiments and collected the data. N.P.B. and R.D. analyzed the data and wrote the paper.

## Competing interests

The authors declare no competing interests.
