## [Peer Review File · Nature Communications]

Reviewers' Comments:

Reviewer #1:

Remarks to the Author:

This study examined responses in V4 during a visual search paradigm in macaques. The paradigm allows the distinction between spatial and feature attention. The study is an extension of previous work of this group related to this experimental paradigm. The citation of previous literature would benefit from a more detailed discussion of the Moore lab FEF stimulation effects on V4 neural responses (FEF is adjacent to the area studied by the authors). The work of Peter Schiller on effects of V4 lesions on visual behaviors is also highly relevant and should be discussed. The major findings of this study are that dorsoventral PFC participates in mediating feature/object attentional effects in V4, maybe similar to the role that FEF has relating to spatial attention. A discussion of the literature on object/feature selectivity in PFC would be useful in this context.

p.6 The authors note that a target that is initially in the affected hemifield may end up in the unaffected hemifield as a result of the free viewing nature of the task. How often did this actually happen in each monkey? Further, what is the fraction of initial saccades that were made to the left/right? What about 2nd / 3rd saccade? On a related note, what was the total fraction of time each animal spent searching the left/right hemisphere, and was this affected by muscimol inactivation? These data are important because they may reveal that inactivating the cortex adjacent to the FEF may affect general directionality of saccadic eye movements. I could not find information relating to where the target was from trial to trial; was randomized across the 20 locations?

p.9: The same time period should be chosen for both monkeys for statistical analysis (either 200 or 250ms).

p.10: The responses to targets and no-share distracters are analyzed. What about share-distracters?

p.12: I am not sure the method of dropping cells is trustworthy. How did the authors arrive at a number of 6 units to be dropped. What if 12 units are dropped? The real question is whether there is a correlation between the factors involved, which should be reported for each of the monkeys.

p.13: For the spatial attention task, it would seem useful to distinguish between spatial locations near and far to the target site. Is the magnitude of the attentional effect dependent on this variable?

Note: some of the figure legends are unreadable. In terms of formatting, some of the information could be condensed (i.e. by showing more data in single panels), while important other data now in the supplementary materials (such as example single unit responses) could be part of the main manuscript.

Reviewer #2:

Remarks to the Author:

In their paper, Bichot et al. sought to test whether a region of prefrontal cortex, the ventral prearcuate area (VPA), contributes to the previously reported modulation by feature-based attention in visual area V4. A previous study from the Desimone lab found that not only does VPA exhibit feature-based activity earlier than the frontal eye field (FEF), but inactivation of VPA during a visual search task diminished feature-attention modulation in FEF. In the current study, VPA was inactivated during the elegant visual search task paradigm as used previously, whilst simultaneously recording neural responses in V4. In the control sessions a significant difference in the firing rate of V4 neurons

was observed when the stimulus in the receptive field matched the searched-for target, versus when a distractor was in the receptive field – so-called feature-based modulation. In contrast, when VPA was deactivated this effect was either diminished or significantly reduced, and a significant negative effect on behaviour was observed. Furthermore, modulation presumed to be by spatial attention (higher activity before a saccade into the receptive field vs. a saccade out) was not significantly different between control and VPA deactivation sessions, suggesting that VPA is involved specifically, be it directly or indirectly, in feature-based modulation of V4 responses.

The authors present an important and exciting extension of their previous work on a potential role of VPA in feature-based attention. The results are overall clearly presented and several important controls included. This study will likely be of broad interest to readers but I have a number of concerns, which the authors should address with additional analyses.

Specific Comments

1. Feature selective modulation

The defining characteristics of feature selective modulation are a) that the modulation is not explained by the spatial locus of attention, which the authors carefully consider, and b), that the modulation is specific with respect to the feature-selectivity of the recorded neuron, which is not considered here. Rather such latter link is assumed given knowledge of feature selectivity in V4 across the population. But previous work, e.g. Treue, Trujillo (1999), Hayden & Gallant (2005), Bichot et al. (2005) showed that modulation by feature-based depending on a neuron's feature selectivity. E.g. attention is positive (enhancement) when a neuron's "preferred" feature is attended while it can be negative (suppressive effect) when attention is directed towards a neuron's non-preferred feature.

If, e.g. the feature-based modulation in V4 during VPA was shifted more towards suppressive effects for non-preferred features, while maintaining the positive modulation for preferred features, this might explain the reduced net effect observed here. Considering the feature-selectivity is therefore important also for the interpretation of the key observed results.

Did the authors measure feature-selectivity to the colors/shapes that were included in this study similar to their measurements of receptive field locations as they did in their Bichot et al. (2005) study? If so, the tuning to these features should be related to the feature-based modulation of each unit.

If feature-selectivity was not measured separately, the responses to the "no-share distractors" should be used for such analysis (the substantial number of trials should allow for such analysis). Whether or not the modulation by feature-based attention was systematically linked to the feature-selectivity of the recorded units needs to be made clear to the readers.

2. Representation of key results and statistical analysis:

While the figures in the main manuscript are limited to spike-density functions the distribution of the results are included in the supplements (S1, S4) giving the readers better insights in the session-by-session effect size and reliability, which is very helpful. These supplementary figures, ideally condensed in a single panel using different colors to differentiate control/VPA deactivation condition should be included in the main paper, either as additional panels in Fig. 3 and 4 or separate figure. Please also give the mean/median contrast ratio, for comparison with such contrast metrics in other studies.

The key statistics in this paper are based on interaction effects of an ANOVA. To verify the robustness of the main findings, the authors should include a non-parametric test (given the shape of the distributions) comparing the distributions of the contrast-ratios in S1, S3, S4 across conditions (control vs VPA deactivation) in each animal. (Note that even if in some cases only a trend would be observed and not all of such additional tests would reach significance as found for the ANOVA, it would

important for the transparency of the reported results to give these additional statistics.)

Additional Comments:

It is intriguing that the authors only find a significant deterioration of behavior for the contralateral visual field although feature attention has repeatedly been shown to act across the entire visual field (e.g. Treue & Trujillo, 1999; Saenz et al, 2002, Bichot et al. (2005), Zhang et al. (2018)). The authors provide a speculation about that in the discussion but this contrast with earlier studies warrants highlighting throughout.

Fig. 2A: please provide details on how the anatomical reconstruction was generated (incl. plane).

What were the criteria to include/exclude control/inactivation sessions?

Reviewer #3:

Remarks to the Author:

This paper reports on an elegant and important set of results in which the authors provide the best evidence to date – the possibly better one being an earlier report by the same group – of the identification of a cortical area responsible for controlling feature-based attention. Using an experimental approach pioneered by the authors several years ago in which monkeys are allowed to freely view an array of objects in search of a match to a stimulus cued at the trial start, the impact of reversible (Muscimol) inactivation of VPA on search and its correlates in visual cortex was tested. As reported previously by the same group, inactivation of VPA selectively reduces performance on the search task. Specifically, the number of saccades required to find the target increases, as well as the time it takes. In contrast, monkeys have no clear impairments in saccades to isolated targets. Next, the authors recorded activity within area V4, where the visual responses of neurons are modulated according to their similarity to the search-for target (as shown in numerous studies, including this group). They record from V4 during sessions both without and during VPA inactivation and compare the ability of V4 activity to distinguish between targets and non-targets. Remarkably, as observed in the activity recorded in two monkeys, the target-dependent modulation is virtually or completely eliminated following VPA inactivation (Figure 3). This is a fantastic result as it means that VPA seems to be a primary source, if not the only source, of feature-specific information during search! Moreover, it shows how that information is used to filter visual information being processed within posterior visual cortex. The authors go on to show (Figure 4) that the loss of modulation is restricted to feature-based modulation as there remains spatially based enhancement when monkeys make saccades to the receptive field (RF) (regardless of what's in it).

The results presented provide a crucial piece to the puzzle of how visual information is filtered by behavioral goals during perception, and is fundamentally important for our understanding of attentional control in the primate brain (which by-the-way is typically carried out over the course of scanning eye movements). Below, I offer some suggestions that I think may clarify the results presented.

1. One immediate concern that often comes up in a design as this in which there is both a behavioral and an neural deficit is how does one know that the latter is not an indirect effect of the former. That is, one must consider the possibility that upon the animal first experiencing an impairment on the search task, it might subsequently employ less effort in carrying it out. As a result, there could be a diminution in search-related modulation in area V4. I would tend to think that such a possibility is quite unlikely to be true, if only because the deficit is both modest, and confined to the contralateral location, thus suggesting that the animal is maintaining a normal level of effort. (Moreover, there is an

increment in performance in the ipsilateral field.) But I wonder if the authors perhaps have other data or reasoning to assuage this concern.

2. Recently the same laboratory published a paper showing that attentional modulation within area V4 is (perhaps critically) dependent upon the incidence of microsaccades toward the neuronal receptive field. This observation raises a couple of important questions in the present data. The first, and less crucial, question is whether or not both types of attentional modulation (spatial and feature-based) observed in this dataset are both dependent on microsaccades in control sessions. The second, more crucial, question is what, if anything, does the inactivation do to the pattern of microsaccades during the search behavior, and whether such changes can account for the loss in modulation.

3. Did the authors also examine the LFPs within V4? If so, are there any observable changes in feature-attention modulation in them following VPA inactivation?

We thank the reviewers for the careful reviews. We have addressed their questions, concerns, and suggestions below. Changes in response to reviewers' comments are highlighted in red in the manuscript.

Reviewer #1 (Remarks to the Author):

This study examined responses in V4 during a visual search paradigm in macaques. The paradigm allows the distinction between spatial and feature attention. The study is an extension of previous work of this group related to this experimental paradigm. The citation of previous literature would benefit from a more detailed discussion of the Moore lab FEF stimulation effects on V4 neural responses (FEF is adjacent to the area studied by the authors). The work of Peter Schiller on effects of V4 lesions on visual behaviors is also highly relevant and should be discussed. The major findings of this study are that dorsoventral PFC participates in mediating feature/object attentional effects in V4, maybe similar to the role that FEF has relating to spatial attention. A discussion of the literature on object/feature selectivity in PFC would be useful in this context.

We have added references and brief discussion of the studies by Moore and by Schiller mentioned by the reviewer, as well as studies of object/feature selectivity in PFC.

p.6 The authors note that a target that is initially in the affected hemifield may end up in the unaffected hemifield as a result of the free viewing nature of the task. How often did this actually happen in each monkey? Further, what is the fraction of initial saccades that were made to the left/right? What about 2nd / 3rd saccade? On a related note, what was the total fraction of time each animal spent searching the left/right hemisphere, and was this affected by muscimol inactivation?

These data are important because they may reveal that inactivating the cortex adjacent to the FEF may affect general directionality of saccadic eye movements. I could not find information relating to where the target was from trial to trial; was randomized across the 20 locations?

During inactivation sessions, the mismatch between target hemifield at the beginning of the trial and target hemifield relative to fixations during free-viewing (i.e., a contralateral or ipsilateral target at the beginning of the trial was contralateral or ipsilateral to fixation after a saccade, respectively) occurred with 18.8% and 18.9% of fixations in monkeys F and J, respectively. We have added this information to the Results.

VPA inactivation did affect general directionality of saccadic eye movements, decreasing the selection likelihood of contralateral stimuli for fixation in favor of ipsilateral stimuli (eye movements in the vertical direction were negligible and amounted to less than 1% of all saccades regardless of monkey or session type). Overall, in VPA inactivation sessions compared to control sessions, there was a decrease of 8.5% in saccades in the contralateral direction for monkey F (52.8% vs. 44.3%; T-test, $t_{10} = 6.2$, $P < 10^{-4}$) and 8.8% for monkey J (47.2% vs. 38.4%; T-test, $t_{12} = 3.6$, $P = 0.004$). This effect was present and significant for the first saccade after array onset as well as subsequent saccades for both monkeys, but it was more prominent for the first saccade compared to the subsequent ones (-24.8% vs. -4.9% and -17.3 vs. -6.2% for monkeys F and J, respectively). As a result of this selection bias in favor of ipsilateral stimuli, the

total fraction of time spent searching the unaffected ipsilateral hemisphere (relative to initial fixation) during inactivation sessions increased by 7.3% and 6.9% compared to control sessions in monkeys F and J, respectively. We have added these findings to the Results.

The information about target locations from trial to trial was previously stated in the Experimental Procedures – Behavioral Tasks section which included the following: “An experimental block of search trials consisted of an animal successfully finding each target color/shape combination at each location once (i.e., 7 colors x 7 shapes x 20 positions = 980 correct trials). The target location and identity were chosen pseudo-randomly from trial to trial such that all desired target identity and target location combinations were completed successfully.”

p.9: The same time period should be chosen for both monkeys for statistical analysis (either 200 or 250ms).

The time periods were chosen for each monkey based on its average reaction time. Given that the reaction times for each monkey were different, we did not feel that a common window for both would be justified. For all analyses, only spikes occurring before saccade initiation were used because the movement of the eyes would cause a movement of the stimulus in the receptive field. Extending the analysis time interval for the monkey with shorter reaction times would decrease the reliability of activity estimates due to increasingly fewer trials contributing to activity at those later “extended” time points. On the other hand, artificially truncating the analysis time interval for the monkey with longer reaction times would yield an incomplete view of cell’s activity developing towards saccade initiation. This would be particularly true in the analyses of attentional effects where one could easily argue that effects developing closer to the time of saccade initiation were being missed. The same reasoning applies to why we use different analysis periods for detection trials vs. search trials. We also note that we did not compare the magnitude of firing rates between the two monkeys.

p.10: The responses to targets and no-share distractors are analyzed. What about share-distractors?

We have added an analysis of same-color and same-shape distractors (Figures S4 and S5). Overall, the enhancement for distractors sharing a target feature was much smaller than for the target. In fact, we found no significant enhancement for same-shape distractors in either control or inactivation sessions, with no interaction between session types for either monkey. In contrast, there was a modest and significant enhancement for same-color distractors during control sessions, and this effect was eliminated during inactivation sessions for both monkeys. Thus, much like the object feature modulation observed for the target in the RF, the color feature modulation observed for a distractor sharing the target color was eliminated after inactivation of VPA.

p.12: I am not sure the method of dropping cells is trustworthy. How did the authors arrive at a number of 6 units to be dropped. What if 12 units are dropped? The real question is whether there is a correlation between the factors involved, which should be reported for each of the monkeys.

To address this concern, we calculated the correlation between the magnitude of detection responses and the magnitude of the feature attention effect in either session type (control or deactivation), and

there was no significant correlation. This is now reported in the Results in the section in question. We also left in the previous analysis, which we feel has merit, although both the results of the old and new analyses are consistent with each other. The way we arrived at the number of 6 units was as follows, “To ensure that the effects of VPA deactivation on responses during feature attention were not caused by overall lower visual responsiveness in this later time interval, we iteratively removed cells with the highest responses in detection trials in control sessions and cells with the lowest responses during detection trials in the inactivation sessions for this monkey until there was no significant difference in overall visual responsiveness during detection trials between the remaining populations of cells (6 cells removed from each population; T-test; $t_{91} = 1.8$, $P = 0.07$).” The number of six cells came from this analysis.

p.13: For the spatial attention task, it would seem useful to distinguish between spatial locations near and far to the target site. Is the magnitude of the attentional effect dependent on this variable?

To address this concern we computed the distance to the target in the different conditions and session types. For both monkeys, there was no difference in the distance to the target between control and inactivation sessions for saccades made to the RF or away from the RF. We have added this information to the Results.

Note: some of the figure legends are unreadable. In terms of formatting, some of the information could be condensed (i.e. by showing more data in single panels), while important other data now in the supplementary materials (such as example single unit responses) could be part of the main manuscript.

We thank the reviewer for pointing out the readability problems. We have increased the font size in the figure legends where it was difficult to read. As suggested by this reviewer and Reviewer #2, we have now added more of the important information from the Supplementary Materials reflecting session-to-session variability (namely the contrast ratio information in cumulative distribution form) to the main figures.

Reviewer #2 (Remarks to the Author):

In their paper, Bichot et al. sought to test whether a region of prefrontal cortex, the ventral prearcuate area (VPA), contributes to the previously reported modulation by feature-based attention in visual area V4. A previous study from the Desimone lab found that not only does VPA exhibit feature-based activity earlier than the frontal eye field (FEF), but inactivation of VPA during a visual search task diminished feature-attention modulation in FEF. In the current study, VPA was inactivated during the elegant visual search task paradigm as used previously, whilst simultaneously recording neural responses in V4. In the control sessions a significant difference in the firing rate of V4 neurons was observed when the stimulus in the receptive field matched the searched-for target, versus when a distractor was in the receptive field – so-called feature-based modulation. In contrast, when VPA was deactivated this effect was either diminished or significantly reduced, and a significant negative effect on behaviour was observed. Furthermore, modulation presumed to be by spatial attention (higher activity before a saccade into the

receptive field vs. a saccade out) was not significantly different between control and VPA deactivation sessions, suggesting that VPA is involved specifically, be it directly or indirectly, in feature-based modulation of V4 responses.

The authors present an important and exciting extension of their previous work on a potential role of VPA in feature-based attention. The results are overall clearly presented and several important controls included. This study will likely be of broad interest to readers but I have a number of concerns, which the authors should address with additional analyses.

Specific Comments

1. Feature selective modulation

The defining characteristics of feature selective modulation are a) that the modulation is not explained by the spatial locus of attention, which the authors carefully consider, and b), that the modulation is specific with respect to the feature-selectivity of the recorded neuron, which is not considered here. Rather such latter link is assumed given knowledge of feature selectivity in V4 across the population. But previous work, e.g. Treue, Trujillo (1999), Hayden & Gallant (2005), Bichot et al. (2005) showed that modulation by feature-based depending on a neuron's feature selectivity. E.g. attention is positive (enhancement) when a neuron's "preferred" feature is attended while it can be negative (suppressive effect) when attention is directed towards a neuron's non-preferred feature. If, e.g. the feature-based modulation in V4 during VPA was shifted more towards suppressive effects for non-preferred features, while maintaining the positive modulation for preferred features, this might explain the reduced net effect observed here. Considering the feature-selectivity is therefore important also for the interpretation of the key observed results.

We should clarify that the data in this and our previous studies (e.g. Bichot et al. 2005) we do not distinguish between facilitation of target locations vs. inhibition of distractor locations. We do not have a stimulus condition comparable to search but viewed passively to establish a baseline response to determine how activity is subsequently modulated when a stimulus becomes a target or a distractor. We referred to a greater response to a stimulus when it is the target than when it is a distractor as enhancement.

Did the authors measure feature-selectivity to the colors/shapes that were included in this study similar to their measurements of receptive field locations as they did in their Bichot et al. (2005) study? If so, the tuning to these features should be related to the feature-based modulation of each unit. If feature-selectivity was not measured separately, the responses to the "no-share distractors" should be used for such analysis (the substantial number of trials should allow for such analysis). Whether or not the modulation by feature-based attention was systematically linked to the feature-selectivity of the recorded units needs to be made clear to the readers.

As the reviewer suggested, we conducted the analysis of the relationship between feature selectivity and feature-based attentional modulation by using responses to the no-share distractors. The responses to the no-share distractors were recorded in the same trials as the responses to the other stimuli, and therefore provided the best estimate of feature selectivity. To quantify the visual selectivity of the neurons throughout the session, we measured average activity in the period of 70-135 ms after array onset during which neurons exhibit a robust evoked response to the no-share distractors that was

not yet modulated by attention (Fig. 3A), and was independent of the period in which attentional modulation was measured (i.e., from 150 ms onwards). In both monkeys, neurons exhibited similar visual selectivity for colors and shapes in control vs. inactivation sessions, calculated as the ratio of the response to the best color/shape and the response to the worst color/shape, with better selectivity for color than shape. We then computed the correlation between the responses to the different colors and shapes in this period to the magnitude of attentional modulation (response to target – response to no-share distractor) when those colors and shapes were the target vs. when they formed a no-share distractor. We found no correlation between visual selectivity and feature-based attentional modulation in either monkey or session type before the first saccade or subsequent saccades, with a general enhancement for the target across all features in control sessions, and no enhancement for the target across any feature in inactivation sessions. Results were the same when we limited the analysis to the most and least preferred colors and shapes. In the previous study cited by the reviewer (Bichot et al, 2005), we found stronger attentional effects for preferred features than non-preferred ones, but the single-neurons recorded in that study were more strongly feature selective than in the current study. Zhou and Desimone (2011) also found comparable attentional enhancement for both preferred and non-preferred features, and the feature selectivity in that study was similar to that in the present study.

2. Representation of key results and statistical analysis:

While the figures in the main manuscript are limited to spike-density functions the distribution of the results are included in the supplements (S1, S4) giving the readers better insights in the session-by-session effect size and reliability, which is very helpful. These supplementary figures, ideally condensed in a single panel using different colors to differentiate control/VPA deactivation condition should be included in the main paper, either as additional panels in Fig. 3 and 4 or separate figure. Please also give the mean/median contrast ratio, for comparison with such contrast metrics in other studies.

The key statistics in this paper are based on interaction effects of an ANOVA. To verify the robustness of the main findings, the authors should include a non-parametric test (given the shape of the distributions) comparing the distributions of the contrast-ratios in S1, S3, S4 across conditions (control vs VPA deactivation) in each animal. (Note that even if in some cases only a trend would be observed and not all of such additional tests would reach significance as found for the ANOVA, it would be important for the transparency of the reported results to give these additional statistics.)

We have followed the reviewer's suggestion and have now included cumulative distributions of contrast ratios in the figures in the main manuscript which are more amenable to showing data from different conditions in the same panel and thus easier to visualize differences. Even though they offer redundant information, we have kept the bar plots in the Supplementary figures in case some readers feel more comfortable viewing the data plotted in that manner. Median contrast ratios are shown in all distribution plots (arrows for cumulative distributions, and dotted lines in the bar plots). As suggested by the reviewer, we have also added non-parametric tests (Wilcoxon rank sum test) comparing the median contrast ratios between experimental conditions. Despite the lower power of non-parametric tests, the statistical significance of the effects of inactivation were confirmed by these tests.

Additional Comments:

It is intriguing that the authors only find a significant deterioration of behavior for the contralateral visual field although feature attention has repeatedly been shown to act across the entire visual field

(e.g. Treue & Trujillo, 1999; Saenz et al, 2002, Bichot et al. (2005), Zhang et al. (2018)). The authors provide a speculation about that in the discussion but this contrast with earlier studies warrants highlighting throughout.

While our observed deficits in the contralateral hemifield coupled with no change (or even improvement) in the ipsilateral hemifield may at first seem at odds with studies (including our own, Bichot et al. 2005) showing that feature attention acts in parallel across the entire visual field, this perceived inconsistency is simply due to the fact that we only unilaterally inactivated VPA. Our results are consistent with our previous findings (Rossi et al. 2007) as well as others' (e.g. Pasternak et al. 2015) showing such contralesional deficits following unilateral PFC lesions in feature selection tasks. In other words, VPA in each hemisphere provides feature bias in the contralateral hemifield. Had we bilaterally inactivated, it is highly likely that we would have observed widespread deficits covering the entire visual field. We have added this clarification to the Discussion.

Fig. 2A: please provide details on how the anatomical reconstruction was generated (incl. plane).

The reconstruction was based on coronal sections from structural MRI images (3T MPRAGE, 500 μ m isotropic). This information has been added to the figure legend.

What were the criteria to include/exclude control/inactivation sessions?

Experimental sessions in which monkeys completed less than half of the required trials in a block (i.e., $1180/2 = 590$) were excluded as they did not provide enough trials for reliable analyses. We have added this information to the Methods section. Two such inactivation sessions occurred with monkey J.

Reviewer #3 (Remarks to the Author):

This paper reports on an elegant and important set of results in which the authors provide the best evidence to date – the possibly better one being an earlier report by the same group – of the identification of a cortical area responsible for controlling feature-based attention. Using an experimental approach pioneered by the authors several years ago in which monkeys are allowed to freely view an array of objects in search of a match to a stimulus cued at the trial start, the impact of reversible (Muscimol) inactivation of VPA on search and its correlates in visual cortex was tested. As reported previously by the same group, inactivation of VPA selectively reduces performance on the search task. Specifically, the number of saccades required to find the target increases, as well as the time it takes. In contrast, monkeys have no clear impairments in saccades to isolated targets. Next, the authors recorded activity within area V4, where the visual responses of neurons are modulated according to their similarity to the search-for target (as shown in numerous studies, including this group). They record from V4 during sessions both without and during VPA inactivation and compare the ability of V4 activity to distinguish between targets and non-targets. Remarkably, as observed in the activity recorded in two monkeys, the target-dependent modulation is virtually or completely eliminated following VPA inactivation (Figure 3). This is a fantastic result as it means that VPA seems to be a primary source, if not the only source, of feature-specific information during search! Moreover, it shows how

that information is used to filter visual information being processed within posterior visual cortex. The authors go on to show (Figure 4) that the loss of modulation is restricted to feature-based modulation as there remains spatially based enhancement when monkeys make saccades to the receptive field (RF) (regardless of what's in it).

The results presented provide a crucial piece to the puzzle of how visual information is filtered by behavioral goals during perception, and is fundamentally important for our understanding of attentional control in the primate brain (which by-the-way is typically carried out over the course of scanning eye movements). Below, I offer some suggestions that I think might clarify the results presented.

1. One immediate concern that often comes up in a design as this in which there is both a behavioral and a neural deficit is how does one know that the latter is not an indirect effect of the former. That is, one must consider the possibility that upon the animal first experiencing an impairment on the search task, it might subsequently employ less effort in carrying it out. As a result, there could be a diminution in search-related modulation in area V4. I would tend to think that such a possibility is quite unlikely to be true, if only because the deficit is both modest, and confined to the contralateral location, thus suggesting that the animal is maintaining a normal level of effort. (Moreover, there is an increment in performance in the ipsilateral field.) But I wonder if the authors perhaps have other data or reasoning to assuage this concern.

We agree with the reviewer that the animals' impairments were confined to one hemifield, and, if anything, it improved in the ipsilateral field following VPA deactivation. In addition, we note that the animal did not know beforehand in which hemifield the target would be located. Both of these facts argue against the possibility that the neural effects were the result of a general reduction in effort. We added a note about this in the Discussion.

2. Recently the same laboratory published a paper showing that attentional modulation within area V4 is (perhaps critically) dependent upon the incidence of microsaccades toward the neuronal receptive field. This observation raises a couple of important questions in the present data. The first, and less crucial, question is whether or not both types of attentional modulation (spatial and feature-based) observed in this dataset are both dependent on microsaccades in control sessions. The second, more crucial, question is what, if anything, does the inactivation do to the pattern of microsaccades during the search behavior, and whether such changes can account for the loss in modulation.

Indeed, a recent study from our lab showed that microsaccades towards an attended stimulus in a covert attention period were associated with an enhanced response to the attended stimulus. These microsaccades occurred around every 300 ms, consistent with other studies. In the present study, the animal made large saccades from one stimulus to another, with a mean interval of only 250ms (median even shorter) between saccades, which did not allow enough time for microsaccades to occur between the larger saccades. In fact, consistent with the idea that microsaccades are simply small saccades, the microsaccades in the previous study and the larger saccades in the present study occurred at roughly similar rates, and in both studies there was an enhancement of response associated with a saccade to an attended stimulus in the receptive field. The enhanced responses associated with a saccade to the receptive field persisted following the VPA deactivation in the present study. We have addressed the

comparison to that study in the Results section describing the overall behavioral performance of the monkeys.

3. Did the authors also examine the LFPs within V4? If so, are there any observable changes in feature-attention modulation in them following VPA inactivation?

Because the LFPs were influenced by the saccades, and the inter-saccade interval was very short, the analysis of LFPs in this interval is beyond the scope of the present study.

Reviewers' Comments:

Reviewer #1:

Remarks to the Author:

I have reviewed the responses of the authors to my questions and i conclude that they have addressed the raised concerns in a satisfactory manner.

Reviewer #2:

Remarks to the Author:

In their revision the authors performed most of the additional analyses and modifications previously requested but my first main concern (point 1) has not been fully addressed, detailed below.

A characteristic of feature-selective attentional modulation in sensory areas is that it can typically be systematically linked to the sensory feature-selectivity of the recorded neuron.

The authors examine whether visual selectivity and feature-based modulation were correlated, and found no correlation. Such lack of correlation is interesting but potentially orthogonal to question of whether the feature-based modulation was systematically associated with stimulus preference of the recorded neuron.

The core comparison is whether the feature-based attentional modulation to a target stimulus of the neuron's preferred stimulus feature exceeded that to a target stimulus of neuron's anti-preferred (null) stimulus (analogous to the findings in Bichot et al. 2005) (cf. supplementary Fig. 6 in Zhou and Desimone, 2011).

The authors potentially refer to such an analysis when they write "Results were the same when we limited the analysis to the most and least preferred colors and shapes." but provide no statistics.

Since the comparison outlined above is central to addressing an important point regarding the potential mechanisms of feature selectivity (cf. Martinez-Trujillo (2011) for a discussion), it needs to be clear that such a comparison is performed (as e.g. done in Zhou and Desimone, 2011), and statistics included.

Reviewer #3:

Remarks to the Author:

The authors have adequately addressed all of my concerns.

Reviewer #1 (Remarks to the Author):

I have reviewed the responses of the authors to my questions and i conclude that they have addressed the raised concerns in a satisfactory manner.

Reviewer #2 (Remarks to the Author):

In their revision the authors performed most of the additional analyses and modifications previously requested but my first main concern (point 1) has not been fully addressed, detailed below.

A characteristic of feature-selective attentional modulation in sensory areas is that it can typically be systematically linked to the sensory feature-selectivity of the recorded neuron. The authors examine whether visual selectivity and feature-based modulation were correlated, and found no correlation. Such lack of correlation is interesting but potentially orthogonal to question of whether the feature-based modulation was systematically associated with stimulus preference of the recorded neuron.

The core comparison is whether the feature-based attentional modulation to a target stimulus of the neuron's preferred stimulus feature exceeded that to a target stimulus of neuron's anti-preferred (null) stimulus (analogous to the findings in Bichot et al. 2005) (cf. supplementary Fig. 6 in Zhou and Desimone, 2011). The authors potentially refer to such an analysis when they write "Results were the same when we limited the analysis to the most and least preferred colors and shapes." but provide no statistics. Since the comparison outlined above is central to addressing an important point regarding the potential mechanisms of feature selectivity (cf. Martinez-Trujillo (2011) for a discussion), it needs to be clear that such a comparison is performed (as e.g. done in Zhou and Desimone, 2011), and statistics included.

We are sorry that we neglected to add the description of statistical tests supporting the statement that the results were the same regardless of whether we used the least preferred or most preferred colors and shapes. These statistics supporting that statement are now added in the text. Changes are highlighted in red in the manuscript.

Reviewer #3 (Remarks to the Author):

The authors have adequately addressed all of my concerns.

Reviewers' Comments:

Reviewer #2:

Remarks to the Author:

I have no additional comments on the manuscript.